# SUBSPACE NODE PRUNING

## ABSTRACT

Efficiency of neural network inference is undeniably important in a time where commercial use of AI models increases daily. Node pruning is the art of removing computational units such as neurons, filters, attention heads, or even entire layers to significantly reduce inference time while retaining network performance. In this work, we propose the projection of unit activations to an orthogonal subspace in which there is no redundant activity and within which we may prune nodes while simultaneously recovering the impact of lost units via linear least squares. We identify that, for effective node pruning, this subspace must be constructed using a triangular transformation matrix, a transformation which is equivalent to and unnormalized Gram-Schmidt orthogonalization. We furthermore show that the order in which units are orthogonalized can be optimised to maximally reduce node activations in our subspace and thereby form a more optimal ranking of nodes. Finally, we leverage these orthogonal subspaces to automatically determine layer-wise pruning ratios based upon the relative scale of node activations in our subspace, equivalent to cumulative variance. Our proposed method reaches state of the art when pruning ImageNet trained VGG-16 and rivals more complex state of the art methods when pruning ResNet-50 networks across a range of pruning ratios.

## 1 INTRODUCTION

With significant progress in the development of neural networks by the research community, commercial interest has recently taken off. Evermore capable models spark private and public interest, excitement, and engagement. However, the computational resources required to train and run these models are immense and neural networks have long exceeded the computational capacity of general purpose hardware.

A variety of approaches have been developed to reduce the computational footprint of models without changing their structure. These range from low-level hardware optimizations (Choquette et al., 2021; Jouppi et al., 2018) to high-level software developments (Paszke et al., 2019; Abadi et al., 2015; Bradbury et al., 2018). Additionally, the representations of models in software have been made more compact with quantization methods (Krishnamoorthi, 2018; Gholami et al., 2022).

More promising are, however, methods which modify and compress neural network models to reduce computational cost while maintaining accuracy. Network compression is possible due to the fact that deep neural networks are found to be significantly over-parameterized in practice, with sometimes orders of magnitude more parameters than should be necessary for computations (Frankle & Carbin, 2018). In this work, we focus on the sub-field of network pruning and develop a new state-of-the-art method within this domain.

### PRUNING NEURAL NETWORKS

The goal of neural network pruning is to reduce the computational execution (inference) time of a model while maintaining its performance. Unstructured approaches which prune the weights of a model result in arbitrarily sparse weight matrices whose multiplication cannot easily be accelerated at compute time – i.e. without translation to real-world inference efficiency. It is therefore desirable to prune whole nodes, convolutional filters, transformer heads, or other structured groups of parameters. Herein, we refer to any of these sub-parts of networks as network 'units'. Thus, the question is posed: how should one choose which units of a network to prune first? Two different approaches emerged

to address this question. First, pruning of pre-trained networks and, second, pruning iteratively while training. While we limit our investigation to the former class that assumes starting out with a well-performing model, the latter class holds great potential for also reducing the cost of training next to the cost of inference.

**Importance scores** When choosing which units of a network to prune, there must first be an attribution of the relative importance of each network unit. The score computed to measure this is known as the importance score. Methods which relate the importance score to the magnitude of weights are prevalent, and these approaches try to minimize the impact of pruning on the network's computation. Li et al. (2016) used the total absolute sum (or squared sum) of incident weights to a convolutional filter as an importance score for that filter. Methods of greater complexity are 'data-driven', making use of training data for forward (and in some cases backward) passes to measure importance scores. Molchanov et al. (2016; 2019) found that the square summed weight-gradient multiplication of weights incident to a node can be used as a theoretically justified importance score under a linear approximation of a network via a 1$^{st}$-order Taylor-expansion (and similarly for the 2$^{nd}$-order case). Theis et al. (2018) extend this notion to using the Fisher information instead, while Yu et al. (2018) proposed a scaling of the weight-magnitude with downstream importance scores to estimate the importance of a unit that minimizes the change in loss induced by pruning.

The above methods aim to assign importance scores such that the smallest importance is assigned to network units which are thought to minimally disturb the network's computation. Other approaches deviate from this assumption. For example, Liu et al. (2023) and Zhang et al. (2022) look at feature maps to determine filters which have greatest task-relevant information. Alternatively, a number of methods also make use of correlation measures of feature maps or filter/node outputs within or between layers, with greater correlations equated to lower importance scores due to the redundancy of these representations (Zhang et al., 2022; Ayinde et al., 2019; Cuadros et al., 2020; Mariet & Sra, 2015; Kim et al., 2020; Goldberg et al., 2022) or a variation thereof (He et al., 2019).

**One-step reconstruction** Some methods go beyond the step of simply removing nodes when pruning and additionally carry out a form of 'one-step reconstruction'. This reconstruction modifies the parameters of a pruned network such that its nodes approximately output the same values which they did prior to pruning. This is unlike retraining or finetuning as it is a single step of modification of the pruned parameters for reconstruction of the pre-pruned outputs. Mariet & Sra (2015) identified redundant nodes by using determinantal point processes, pruned these nodes, and recovered their impact on a network by linear least squares (LLS) approximation between the original and pruned pre-activations. Similarly, He et al. (2017) used LLS to approximate pruned nodes, while selecting nodes based on LASSO regression.

In contemporary work, Luo et al. (2017) estimated a single scalar value per node to best reconstruct the lost activity, a method with lower expressiveness than the aforementioned methods. Kim et al. (2020); Goldberg et al. (2022) chose yet a different, data-free, approach to reconstructing unit activity by moving weight parameters between layers, however this was only applicable to networks with rectified linear unit (ReLU) non-linearities. Chin et al. (2018) went further still, into a non-linear least squares solution attempt using evolutionary algorithms. Similar methods for reconstructing unit activity for large language models have recently been presented (Li et al., 2024; Frantar & Alistarh, 2023).

**Global importance** Despite the importance of local scoring (i.e. scoring of units within a layer), there needs to be a notion of unit importance across layers, a so called 'global importance'. While for some approaches such global ranking is a natural consequence of the local importance estimation (Molchanov et al., 2016; 2019; Yu et al., 2018), several methods only provided local importances and rely on expert knowledge, manual exploration(Wang et al., 2021), or simple assumptions such as the equivalence of pruning at any layers Li et al. (2016). The most advanced methods rely on measurement of some form of network 'sensitivity' to achieve peak performance(You et al., 2019).

**This work** We propose and demonstrate the efficacy of three novel ideas for improved node pruning. First, a one-step reconstruction method which relies upon the construction of a subspace in which unit activations are factorized and made orthogonal. In this subspace, nodes can be pruned with immediate reconstruction of layer outputs by LLS. Second, a novel importance scoring method based only upon non-redundant unit activities. Third, the measurement of global unit importance based

upon the percent of variance explained within our proposed subspace. Figure 1 illustrates these three contributions in order.

## 2 SUBSPACE NODE PRUNING (SNP)

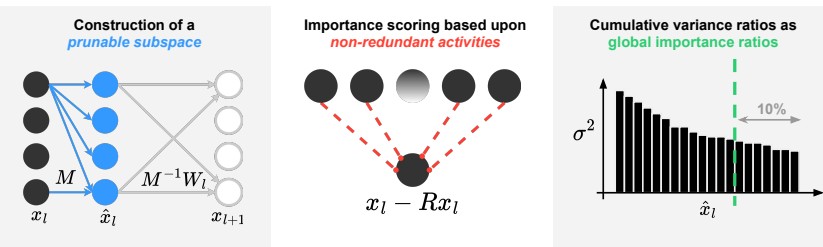

Figure 1: Graphical depiction of our main methods and contributions. Left: Our subspace construction method involves the projection of unit activities to a subspace via a lower-triangular matrix transform. This induces an intermediate 'latent' space of orthogonal activity vectors. Centre: We propose a manner to isolate non-redundant activities by the measurement and removal of any linearly decodable information from all other units in a layer. Right: We can interpret the variances of the processed activities in our subspace as their importance and prune based upon a global variance-based cutoff, rather than by determining specific numbers of units to remove at each layer.

### 2.1 FACTORISING NEURAL CONTRIBUTIONS

Consider a typical deep neural network (DNN) architecture, in which the outputs at each layer, $l \in \{1, \ldots, L\}$, are defined as $\mathbf{X}_l = f_l(\mathbf{Y}_l) = f_l(\mathbf{W}_{l-1}\mathbf{X}_{l-1})$, where $\mathbf{X}_l \in \mathbb{R}^{n_l \times s}$ is a tensor of outputs for layer $l$ consisting of $n_l$ units for $s$ samples. These layer outputs are composed based upon a matrix multiplication of the previous layer outputs, from weights $\mathbf{W}_l \in \mathbb{R}^{n_{l+1} \times n_l}$, and by an element-wise transfer function $f_l(\cdot)$. We consider these fully-connected deep neural networks to introduce our approach, however, we shall also treat convolution in the results that follow.

In a pruning pipeline, assuming that importance scores were already available, the next step would be to prune $m$ input units and their associated weight vectors, starting with the unit of lowest importance score. Here we instead propose an intermediate step. We propose that one might remove as much redundant activity from units in a given layer to ensure that pruning has the least possible impact on network dynamics.

In the interest of doing so, consider the pre-activations $\mathbf{Y}_{l+1} = \mathbf{W}_l\mathbf{X}_l \in \mathbb{R}^{n_{l+1} \times s}$ at layer $l + 1$ of a network in response to the set of inputs (training dataset). Without specifying the transformation matrix $\mathbf{M}_l$ yet, we can equivalently parameterize a layer $l$ as

$$\mathbf{Y}_{l+1} = \mathbf{W}_l\mathbf{M}_l^{-1}\mathbf{M}_l\mathbf{X}_l = \mathbf{W}_l\mathbf{M}_l^{-1}\hat{\mathbf{X}}_l,$$

where $\hat{\mathbf{X}}_l = \mathbf{M}_l\mathbf{X}_l$ are the original inputs projected into a subspace where there is no redundancy, a space in which activations are orthogonal latent variables. For an unpruned network, the subspace transformation has no impact on the network computation due to our inclusion of the inverse subspace transform, $\mathbf{M}_l^{-1}$. Pruning within the subspace affects the matrices as follows: we prune the columns of $\mathbf{M}_l^{-1}$, and the rows of $\mathbf{M}_l$. While the shape of the matrix product of the two matrices remains unaffected, the product will no longer be the identity matrix, unless unit activity was already orthogonal prior to factorization.

Here, we propose the application of an unnormalized Gram-Schmidt (GS) orthogonalization for the orthogonalizing transformation, via a lower triangular matrix $\mathbf{M}_l$, to find a subspace in which the units have no redundancy in their activities. Specifically, one can frame our desired transformation as one in which we project our unit activity at some layer $l$ to a subspace via a linear projection matrix $\hat{\mathbf{X}}_l = \mathbf{M}_l\mathbf{X}_l$ with the restriction that we wish for the final dot-product between each pair of vectors to be zero (orthogonalized) and for the orthogonalizing matrix $\mathbf{M}_l$ to have a lower triangular structure such that

$$\hat{\mathbf{X}}_l\hat{\mathbf{X}}_l^\top := \mathrm{diag}(\hat{\mathbf{X}}_l\hat{\mathbf{X}}_l^\top) = \mathbf{M}_l\mathbf{X}_l\mathbf{X}_l^\top\mathbf{M}_l^\top .$$

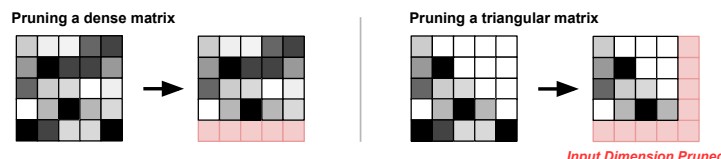

Figure 2: Our choice of a subspace which is constructed for lower-triangular matrices is here justified. Left: If a dense matrix is used to form a subspace, pruning in the dense transformation matrix does not prune the original input nodes. Right: When pruning a lower-triangular transformation matrix, pruning the bottom row corresponds to pruning away an entire input node.

Orthogonalization is possible in multiple ways, for example by principle component analysis (PCA), zero-phase component analysis (ZCA) or otherwise. However, we specifically desire for our orthogonalizing transformation matrix, $\mathbf{M}_l$ to be lower-triangular. In order to understand why, consider two aspects. First, a lower-triangular orthogonalizing matrix means that our units are treated as if they have been ordered by priority, with the first unit orthogonalizing the $n_l - 1$ remaining units, the second unit orthogonalizing the $n_l - 2$ remaining units and so on. This ensures that the final units of the layer have had all possible activity which could be explained by earlier units removed - i.e. that the final units in a layer have had all of the information which could be extracted from other units subtracted away. Second, this lower-triangular setup allows one to prune the latent variables whilst also pruning the original input nodes. This prunability is a natural consequence of the zeros in the upper-triangular section of our transformation matrix, as illustrated in Figure 2, that lead to removing columns through pruning rows from $\mathbf{M}_l$. Hence, pruning in such subspace ultimately results in reduced matrix dimensionality that ultimately prunes the corresponding input node. Note that pruning in alternative subspaces, such as the PCA subspace, does not result in a reduced matrix dimensionality but instead in a low-rank matrix.

Returning to our problem setup, we therefore wish to solve for a lower-triangular orthogonalizing matrix $\mathbf{M}_l$. This can be accomplished by LDL-decomposition based on a rearrangement of the previous equation

$$\mathbf{X}_l{\mathbf{X}_l}^\top = \mathbf{M}_l^{-1}\mathbf{D}_l\left(\mathbf{M}_l^{-1}\right)^\top ,$$

where $\mathbf{D}_l = \mathrm{diag}(\hat{\mathbf{X}}_l\hat{\mathbf{X}}_l^\top)$, the variances of our latent variables.

Given our prior setup, our newly determined transformation matrix provides us with a route to a ranked, orthogonal subspace. In the new subspace, we can now prune from most- to least 'restricted' latent variable (restricted by the triangular structure of the matrix $\mathbf{M}_l$). Therefore, we prune $\mathbf{M}_l$ by removing rows from the bottom. If we denote $\cdot^*$ as pruning the last rows, and $\cdot_*$ as pruning the last columns of a matrix, we reparameterize the weights as

$$\hat{\mathbf{W}}_l = \mathbf{W}_l(\mathbf{M}_l^{-1})_*(\mathbf{M}_l)_*^* .$$

We provide the pseudocode to prune a single layer using the proposed method in Algorithm 1. Note that a similar algorithm can be used to prune entire filters in convolutional networks.

---

**Algorithm 1:** Layer-wise subspace node pruning

---

**Input:** Data $\mathbf{X}_l$, Weights $\mathbf{W}_l$, Number of units to prune $n$
**Output:** Pruned weights $\hat{\mathbf{W}}_l$
$\mathbf{C}_l = \mathbf{X}_l\mathbf{X}_l^T$                        ▷ Compute dot-product between input feature vectors
$\mathbf{M}_l^{-1}, \mathbf{D}_l = \mathrm{LDL}(\mathbf{C}_l)$                         ▷ Decompose matrix $\mathbf{C}_l$
$\hat{\mathbf{W}}_l = \mathbf{W}_l(\mathbf{M}_l^{-1})_{:,:n}(\mathbf{M}_l)_{:n,:n}$         ▷ Prune $\mathbf{M}_l$ and $\mathbf{M}_l^{-1}$ (leading to pruned $\mathbf{W}_l$)
**Return:** $\hat{\mathbf{W}}_l$

---

It can be proven that pruning in this subspace automatically reconstructs the output of such a layer by linear least squares, see Appendix A. This equivalence demonstrates the optimality of our choice of subspace and highlights that via LLS one recovers input activity that is redundant within the set of remaining units. Further, we can therefore also posit that any importance scoring method applied

outside of this subspace shall overestimate the importance of unit activities due to the existence of unaccounted redundancies. We address this overestimation in the next section.

## 2.2 Importance scoring: Reordering units prior to factorization

As mentioned previously, our method of factorising nodes in the GS subspace holds promise for ensuring that the units removed (in the subspace) have minimal activations and that all redundant information is automatically reconstructed. However, we so far have not addressed the question of how one should choose the order in which units are orthogonalized. In fact, we so far considered a GS subspace transformation based upon the default ordering of units in a layer, an ordering which could be much improved.

Generally, the choice of unit ordering is free for a practitioner since it simply changes the order of units from which we compute the GS subspace (consider that one could permute the matrix $\mathbf{M}_l$ so long as you also unpermute via matrix $\mathbf{M}_l^{-1}$). See Appendix B for the pseudo-code of this permutation for pruning individual layers in a network. Some orderings are, however, evidently better than others. Importance scores, as computed by alternative existing work, are good first candidates for a reordering process, allowing the combination of our subspace method with any existing importance scoring method. However, as noted before, these methods include recoverable unit activity in their estimations. In the following, we propose a method to solve this problem at any individual layer.

First, remembering that our subspace construction method aims to orthogonalize units in an ordered fashion, one can ask: which units are best orthogonalized by all other units in a layer? To calculate this, we can attempt to measure how much of each unit's activity variance can be reduced based upon a linear readout from all other units in a layer.

Assuming that one wishes to find a dense matrix, $\mathbf{R}_l$, which computes the amount of redundant activity that each unit has relative to each other unit, we are looking to measure $\hat{\mathbf{X}}_l = \mathbf{X}_l - \mathbf{R}_l \mathbf{X}_l$ where the desired outcome is that $\hat{\mathbf{X}}_l \hat{\mathbf{X}}_l^\top$ is some diagonal matrix of variances, and the diagonal of the matrix $\mathbf{R}_l$ is zeros (i.e. there is no boosting up of a unit's variance).

To solve for this, we can formalize the desired property that $\hat{\mathbf{X}}_l \hat{\mathbf{X}}_l^\top = \mathrm{diag}(\hat{\mathbf{X}}_l \hat{\mathbf{X}}_l^\top) = \mathbf{S}_l^2$ and solve for the value of $\mathbf{S}_l$. To do so, we can compute, $\hat{\mathbf{X}}_l \hat{\mathbf{X}}_l^\top = (\mathbf{I}_l - \mathbf{R}_l)\mathbf{X}_l \mathbf{X}_l^\top(\mathbf{I}_l - \mathbf{R}_l)^\top = \mathbf{S}_l^2$, and therefore

$$\mathbf{X}_l \mathbf{X}_l = (\mathbf{I}_l - \mathbf{R}_l)^{-1}\mathbf{S}_l\mathbf{S}_l^\top(\mathbf{I}_l - \mathbf{R}_l)^{-1^\top}.$$

One may assume that our transformation matrix is symmetric (since the degree to which one node orthogonalizes another is symmetric), and thus $\mathbf{S}_l^{-1}(\mathbf{I}_l - \mathbf{R}_l) = (\mathbf{X}_l \mathbf{X}_l^\top)^{-1/2}$. The transformation is equivalent to the well known ZCA transform (Krizhevsky et al., 2009) though we have now added a term representing the scaling applied to reach a whitened state. To find $\mathbf{S}_l$ note that $\mathrm{diag}(\mathbf{I}_l - \mathbf{R}_l) = \mathbf{I}_l$, as defined earlier, and therefore

$$\mathbf{S}_l = \mathrm{diag}((\mathbf{X}_l \mathbf{X}_l^\top)^{-1/2})^{-1}.$$

The (diagonal) values of the matrix $\mathbf{S}_l$ are the novel importance scores which we propose in this work and we refer to this as the 'unnormalized-ZCA' ordering. Concretely, these values are the L2-norms of each units' activation after each unit has individually been orthogonalized by all other units. This effectively means that it is the scale of each units' activation which is truly unique (non-redundant) with respect to all other unit activations. Assuming centred data, these values are equivalent to the standard deviations of each unit.

It is also possible to generalize the idea and to combine this measure with other existing importance scoring methods in order to discount redundant information when measuring importance. We briefly describe such an extension in Appendix C but do not explore it any further in this work.

## 2.3 Cumulative variances: From pruning layers to pruning networks

In the previous section, we described the measurement of an importance score based upon the remaining norm of a unit's activity after it has been orthogonalized by all other units within a layer. However, measuring a local layer-wise importance score is only sufficient for determining how much

one might prune a single layer. When pruning a whole network, one must then make a determination of how much to prune individual layers.

To address this, we build on the fact that our pruning method yields the activity variances of the latent variables (subspace units) without any extra computation (the diagonal matrix $\mathbf{D}_l$ from LDL-decomposition). With these variances computed, we propose to measure the global importance as the cumulative variance of a unit and all succeeding units, normalized by the total variance of the layer. Mathematically speaking, this means that the global variance-based importance score for a given unit $i$ in layer $l$ is given by

$$\text{Importance}_l^i = \frac{\sum_{j=i}^{n_l} \mathbf{D}_l^{jj}}{\sum_{k=0}^{n_l} \mathbf{D}_l^{kk}}$$

where the repeated superscripts indicate selection of diagonal elements of our $\boldsymbol{D}_l$ matrices. This construction allows us to set a single global parameter (the percent variance to be removed from all layers) which automatically arrives at an individual layer-wise pruning ratio.

## 3 EXPERIMENTS

We demonstrate the efficacy of our proposed method by application to VGG-11, 16, and 19 as well as ResNet-50 architectures. We use the networks from PyTorch (Paszke et al., 2019) pre-trained on the ILSVRC (ImageNet) dataset. The dataset contains 1,281,167 labeled training images and 50,000 labeled validation images. They are split into 1000 object categories that the models try to predict. When measuring the cross-correlation matrix of every layer's activations (inputs), we make use of the full training set images transformed via the test-transforms detailed in Appendix D.

To evaluate model performance, we show results in two cases. The first case considers VGG networks before any retraining. We compare our method against reimplementations of the method of Li et al. (2016), which uses the sum of absolute weights (SAW), as well as the unstructured absolute weight magnitude pruning method (Han et al., 2015). Further, we reimplement and compare against ThiNet (Luo et al., 2017) and PFA-EN (Cuadros et al., 2020). Due to limited computational resources and the cost of performing pruning across these baselines, we make re-implemented comparisons of these baseline method on VGG networks only. The SAW, AW, and ThiNet methods provide no guidance on the global ranking of units, instead assuming that a practitioner might uniformly prune all layers by the same amount. Therefore we employ a uniform pruning ratio across layers when implementing these methods. This means that all layers are pruned by the same ratio of units, referred to as a uniform pruning (uni) in all relevant figures. In contrast, PFA-EN performs PCA to decide a global ranking of units on top of their local importance structure. Notably we apply PFA-EN's PCA measure at input nodes, rather than output activations, finding that this produces best performance. We re-implemented all baselines due to unavailable code or outdated packages and have code available for reproduction of all experiments at <SEE ATTACHED ZIP>.

We furthermore compare our method with three variations in importance scoring. We compare against random importance measurement (SNP-random), the SAW importance measure (SNP-SAW), and our proposed unnormalized-ZCA ranking (SNP-ZCA). Here, SNP refers to the subspace construction and pruning therein. We further combine with our global importance measurement via cumulative variance estimation (var). The exact pruning ratios for our experiments can be found in Appendix E.

The second part of our results investigates performance after retraining. We evaluate the efficacy of our methods on VGG-16 and ResNet-50. For these experiments, we use the retraining recipes detailed in Appendix D. We compare the SNP-SAW/ZCA methods against a selection of baselines copied from literature. So far, we have only described pruning of single-branch networks such as VGG networks. Dealing with networks with multiple branches such as ResNets is not trivial. We show how to prune multi-branch networks in Appendix F.

Due to the prohibitive computational cost involved for retraining all these methods, we retrain only SNP-ZCA var on three different seeds to generate an estimate of the variation in performance. Since all methods are deterministic, however, variation is only introduced based upon the shuffling of data presentation during training.

We report the performance measured as Top-1 test accuracy of the pruned networks in terms of parameter count, FLOPs, runtime and energy consumption. The parameter count and number of

FLOPs are measured using the fvcore package (https://github.com/facebookresearch/fvcore). A FLOP is counted as a multiply-add operation. All performance evaluations are run on 16 threads on Intel(R) Xeon(R) Gold 5218 CPU @ 2.30GHz CPUs and a Quadro RTX 6000 GPU on a compute cluster. The retraining is performed on a faster compute cluster with a Nvidia A100 GPU and 18 CPU cores of Intel Xeon Platinum 8360Y processors.

# 4 RESULTS

### PRUNING WITHOUT RETRAINING

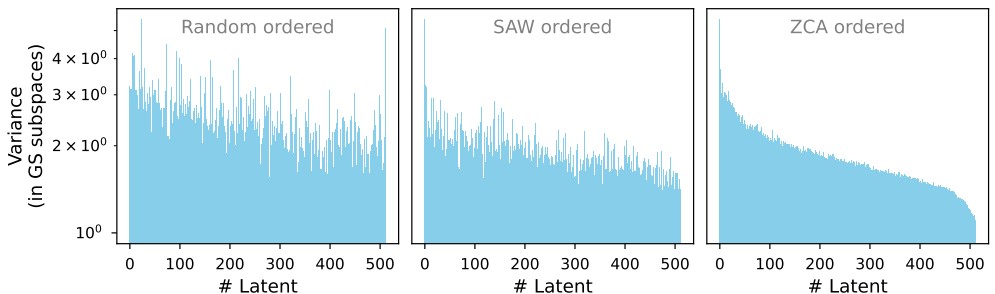

Figure 3: Latent unit variances in our subspace after Gram-Schmidt orthogonalization of layer 12 from VGG-16. Prior to the orthogonalization, the units are ordered either randomly (left), ordered using the SAW importance measure (middle) and ordered using out novel proposed ordering by unnormalized-ZCA (right) variances.

We start our analysis of VGG net pruning by assessing and verifying the efficacy of the proposed reordering strategies. Figure 3 shows the variances of the latent variables, our network units after they have been orthogonalization by our proposed (unnormalized) Gram-Schmidt method. We see that a random ordering of units gives highly noisy variances across a layer. If we permute the unit order according to the importance scores of SAW (prior to GS orthogonalization) the variances remain noisy, but a stronger ranking across units emerges. This indicates that the choice of unit ordering given by the SAW method does measurably help to extract units with the most redundant activities (those which end up with low variance on the far right). In contrast, re-ordering according to our unnormalized-ZCA importance scoring method leads to a very smooth ranking of variances. We hypothesise that this ought to be attributed with better performance since the better units are approximated using preceding units, the lower the approximation error during pruning. We test this hypothesis by application of our method on VGG-11/16 and 19.

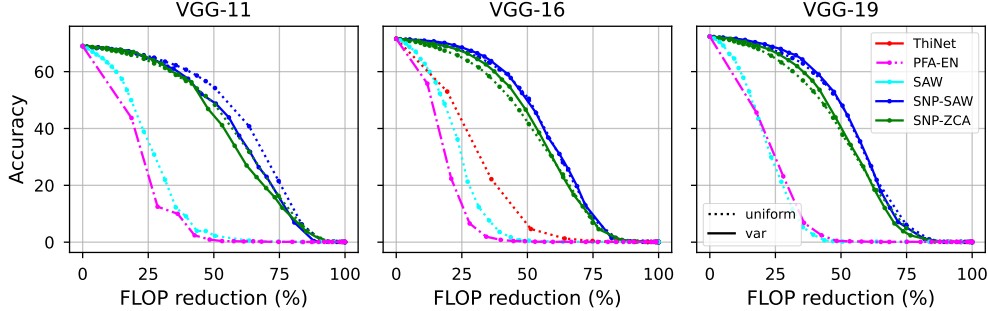

Figure 4: Pruning performance of VGG-11, 16, 19 networks without retraining. Test accuracy against reduction in FLOPs. We compare baseline pruning methods to both our proposed method (SNP-ZCA) and to the combination of our subspace-construction method with the SAW importance ordering (SNP-SAW). Furthermore, we show results both for uniform pruning per-layer (dashed lines) and our proposed global variance-based pruning (solid lines).

Figure 4 shows post-pruning accuracies of the VGG-11/16/19 networks using both our proposed method along with various baselines. In these plots, we also compare our proposed global variance-based pruning determination (labelled 'var') against a uniform pruning process which prunes the same ratio of units at every network layer.

As one can observe, the subspace methods (SNP-SAW and SNP-ZCA) are the most performant, retaining much of the initial performance with a degradation at greater pruning levels. Alternative methods suffer to a much greater degree with significant reductions in test accuracy with even small FLOP reduction. However, the inclusion of a global variance-based cutoff for determining does not show major benefits before retraining. The question remains as to which methods will perform best once these networks are retrained to regain performance post-pruning.

### RE-TRAINING AFTER PRUNING

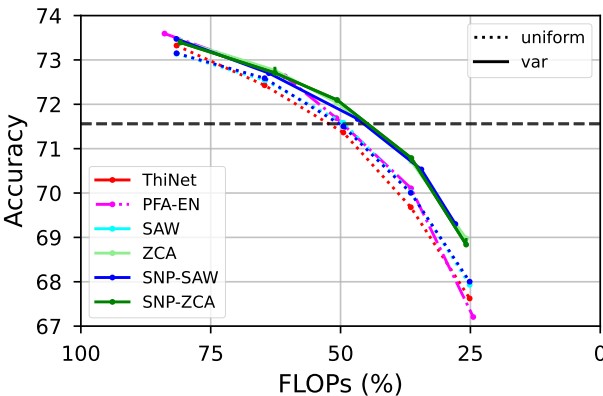

Figure 5: A comparison of our reconstruction method indicated by 'SNP' with our global variance cutoff based importance (var) vs baseline methods. We show VGG-16 networks Top-1 accuracy after retraining against the amount of retained FLOPs. The black strided horizontal line shows the initial network performance before pruning. SNP-ZCA has error bars on top of the datapoints from three randomly seeded training runs, though these are barely distinguishable. Dotted lines refer to using the same pruning ratio for all layers, whereas solid lines refer to our global variance-based importance scoring. PFA-EN is the only unique method which uses PCA to determine global importance.

Figure 5 shows the performance comparison of our models and the baselines after retraining. Table 1 shows final accuracy of the tested methods for various pruning ratios. Most crucially, our global variance-based cutoff (var) for automated layer-wise pruning shows significant performance gain over uniform pruning for FLOP reduction of $2\times$ and greater, demonstrating its efficacy and suitability for network-wide pruning. We see that our SNP-ZCA and SNP-SAW are very similar in performance and outperform all other methods by an increasingly wide margin as we increase the FLOP reduction.

We include also an example of our proposed ZCA-based importance scoring without the subspace reconstruction (see ZCA var), as well as the SAW baseline with subspace reconstruction (see SNP-SAW uniform). These results show that our reconstruction has a measurable but small impact when retraining is done to convergence. This suggests that subspace-based reconstruction might be most useful in cases where retraining is prohibitively expensive or when a marginal gain in retraining speed is desired.

All methods apart from ThiNet are deterministic. We therefore assume little variation between different retraining runs. To verify this, we retrain the SNP-ZCA method with three random seeds and see very little deviation in the performance (shown by the almost invisible error bars). ThiNet is non-deterministic due to its sampling process when collecting data for activity reconstruction.

Regarding our baselines, we find that PFA-EN initially is at an advantage over the other baselines, but is caught up by SAW after pruning roughly 50% of FLOPs ($2\times$ FLOP speedup). The initial high performance of PFA-EN is competitive with our SNP-ZCA and SNP-SAW for the first two pruning

Table 1: **VGG-16 (ImageNet)**: The final accuracies, FLOP count, and Parameter count for our method along with re-implemented alternative methods applied to the VGG-16 network. See Figure 4 for these results in training curve form. To ensure a fair comparison of our results and those in the literature, we group the results into blocks with similar FLOP speed-ups. Each of these blocks naturally has results which are obtained by pruning at different ratios. Groups are separated by dashed lines.

| Importance | Final %Acc | $\Delta$Acc | FLOP speedup | #Params |
|---|---|---|---|---|
| ThiNet (reimpl.) | 71.36 | -0.23 | 2.02× | **68.79M** |
| PFA-EN (reimpl.) | 71.69 | 0.10 | 1.97× | 76.97M |
| SAW (reimpl.) | 71.58 | -0.01 | 2.02× | **68.79M** |
| **SNP-SAW uniform** (ours) | 71.50 | -0.09 | 2.02× | **68.79M** |
| **ZCA var** (ours) | 72.03 | 0.44 | 1.97× | 83.89M |
| **SNP-SAW var** (ours) | 71.67 | 0.08 | **2.14×** | 75.58M |
| **SNP-ZCA var** (ours) | **72.08** ($\pm$0.02) | **0.49** | 1.97× | 83.89M |
| ThiNet (reimpl.) | 69.68 | -1.91 | 2.74× | **50.93M** |
| PFA-EN (reimpl.) | 70.11 | -1.48 | 2.75× | 53.58M |
| SAW (reimpl.) | 70.02 | -1.57 | 2.74× | **50.93M** |
| **SNP-SAW uniform** (ours) | 70.00 | -1.59 | 2.74× | **50.93M** |
| **ZCA var** (ours) | 70.71 | -0.88 | 2.75× | 63.05M |
| **SNP-SAW var** (ours) | 70.53 | -1.06 | **2.90×** | 57.57M |
| **SNP-ZCA var** (ours) | **70.77** ($\pm$0.06) | **-0.82** | 2.75× | 63.05M |

ratios, but then suffers from the stark drop-off. ThiNet is largely outperformed by the other methods under this training recipe, see Appendix D.

Lastly, it is notable that all models far outperform the pre-trained model for small amounts of pruning. This is in line with existing work and shows that pruning in small amounts can act as a regularization to improve the model's generalization performance.

Table 2: **ResNet-50 (ImageNet)**: The final accuracies, FLOP speed-up, and Parameter count for our method along with referenced alternatives from the literature. 'Regular Retraining' indicates whether methods use a regular post-pruning training cycle or if pruning is performed *during* retraining. To ensure a fair comparison of our results and those in the literature, we group the results into blocks with similar FLOP speed-ups. Each of these blocks naturally has results which are obtained by pruning at different ratios. Groups are separated by dashed lines.

| Importance | Pre-prune %Acc | Final %Acc | $\Delta$Acc | FLOP speedup | #Params | Regular Retraining |
|---|---|---|---|---|---|---|
| FPGM(He et al., 2019) | 76.15 | 74.13 | -2.02 | 2.13× | - | ✓ |
| SFP (He et al., 2018) | 76.15 | 74.61 | -1.54 | 1.72× | - | ✗ |
| Taylor-FO (Molchanov et al., 2019) | 76.18 | 74.50 | -1.68 | 1.82× | 14.2M | ✗ |
| SAW (Wang et al., 2023a) | 76.13 | 75.24 | -0.89 | 2.31× | - | ✓ |
| GReg-2 (Wang et al., 2021) | 76.13 | 75.36 | -0.77 | 2.31× | - | ✗ |
| TPP(Wang & Fu, 2023) | 76.13 | **75.60** | **-0.53** | 2.31× | - | ✗ |
| **SNP-ZCA uniform** (ours) | 76.13 | 75.18 | -0.95 | **2.34×** | 11.24M | ✓ |
| **SNP-ZCA var** (ours) | 76.13 | 75.43 | -0.70 | 2.30× | 13.76M | ✓ |
| LFPC (He et al., 2020) | 76.15 | 74.46 | -1.69 | 2.55× | - | ✗ |
| SAW (Wang et al., 2023a) | 76.13 | 74.77 | -1.36 | 2.56× | - | ✓ |
| GReg-2 (Wang et al., 2021) | 76.13 | 74.93 | -1.20 | 2.56× | - | ✗ |
| TPP (Wang & Fu, 2023) | 76.13 | **75.12** | **-1.01** | 2.61× | - | ✗ |
| **SNP-ZCA uniform** (ours) | 76.13 | 74.67 | -1.46 | **2.63×** | 10.00M | ✓ |
| **SNP-ZCA var** (ours) | 76.13 | 75.08 | -1.05 | 2.60× | 12.37M | ✓ |
| Taylor-FO (Molchanov et al., 2019) | 76.18 | 71.69 | -4.49 | 3.05× | 7.9M | ✗ |
| SAW (reimpl.) | 76.13 | 74.13 | -2.00 | 3.03× | 8.77M | ✓ |
| GReg-2 (Wang et al., 2021) | 76.13 | 73.90 | -2.23 | 3.06× | - | ✗ |
| TPP (Wang & Fu, 2023) | 76.13 | **74.51** | **-1.62** | **3.06×** | - | ✗ |
| **SNP-ZCA uniform** (ours) | 76.13 | 74.36 | -1.77 | 3.03× | 8.77M | ✓ |
| **SNP-ZCA var** (ours) | 76.13 | 74.43 | -1.70 | 3.04× | 10.74M | ✓ |

EXTENDING RESULTS TO RESNET-50

Having demonstrated the benefit of our method in VGG network pruning, along with re-implemented baselines we further prune ResNet-50 architectures and compare against reported accuracies achieved in the literature. As before, a pre-trained model is loaded, the training dataset used to compute activity correlations at all layers of the network and thereafter subspace construction, unnormalized-ZCA importance measurement, and global variance-based cutoffs used for layer-wise pruning. The retraining recipe used is given in Appendix D. Note, that as in the previous simulations, our pruning method first prunes and then re-trains with no further pruning during retraining.

Table 2 shows the results of applying our proposed pruning method to the ILSVRC (ImageNet) trained ResNet-50 architecture. As can be seen, our method is highly competitive with existing methods in the range of 2-3× FLOP reduction, consistently in second place and only being outperformed by TPP (Wang & Fu, 2023). We see the performances achieved as a significant accomplishment considering that almost all competitive methods not only prune networks but also carry out some pruning *during* retraining in order to more carefully remove nodes (including TPP). Furthermore, most alternative methods have fixed layer-wise pruning ratios, determined heuristically or otherwise, unlike ours which has a single global variance-based cutoff which automatically determines unique layer-wise pruning ratios.

## 5 DISCUSSION

In this work, we introduced a novel, often state of the art, method for structured pruning of pre-trained deep networks with one-step reconstruction and focus upon the non-redundant variance of each unit's activation. Research into such approaches holds promise for the potential future of pruning without considering redundant activity within a neural network. However, a number of additional areas of exploration remain open, including the conditions in which retraining can be ignored, how to best treat non-redundant unit activations, as well as how to apply such methods *during* training.

On the first note, we find our proposed method to be somewhat successful when pruning networks without retraining. However, to reach competitive performance, retraining is nonetheless essential when significantly pruning networks. This finding is in contrast with recent work which attempted a related reconstruction method for LLMs (Li et al., 2024), where it was claimed that retraining could be unnecessary. Future work should consider determining how significantly our findings apply to such alternative model types.

Second, we make use of unit activities within our subspace to compute layer-wise pruning ratios based upon a global variance-based cutoff. However, this does not at all consider the downstream sensitivity of a network to these activations, implicitly assuming that all unit variances are equally useful. Combining our measure with unit-wise sensitivity measures has the potential to further boost performance. Furthermore, alternative importance measures which focus on non-redundant neural contributions (within our proposed subspace) may hold significant potential, by avoiding allocation of importance to redundant components unit activities.

Finally, we restricted ourselves in this work to examining the pruning of pre-trained networks. A promising extension of this work would be to additionally prune networks during training and retraining. The most competitive methods which we could identify all make use of such a *during* (re)training adjustment of pruned units. Our method could also benefit from such a treatment by maintenance of an orthogonalizaed subspace during training and a more gradual pruning arrangement. This could force a network to encode information in a ranked fashion during training and potentially enable far greater final performance.

In conclusion, we see the subspace node pruning method described herein as a new perspective on the problem of pruning. This method can be combined with any existing method for node selection and importance scoring and has the potential to significantly improve existing node pruning methods. However, there also remain some wide open areas to be explored, including the application to alternative models and during training of models from scratch. Beyond the models investigated here, it also remains to be seen how and whether pruning approaches can truly impact the most energy-expensive of models, such as generative pre-trained transformer models which are currently in vogue.

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

## A    RELATION TO LINEAR LEAST SQUARES

So far, we have introduced a method where pruning latent variables within a GS subspace corresponds to pruning unit inputs. This approach is based on the assumption that pruning within the subspace has minimal impact on the subsequent layer, without any notion of what 'minimal impact' means. In the following, we seek to demonstrate the efficacy of this subspace pruning by proving its equivalence to LLS approximation for recovering the original inputs from their pruned counterparts. Thereby, we show that 'minimally impacting' means to minimize the sum-squared difference of the unit inputs from their pruned counterparts. For clarity, we omit the layer subscript in the following, as the computation is fully contained within each layer.

We start by defining the recovery matrix $\mathbf{A}$ as

$$\arg\min_{\mathbf{A}}\|\mathbf{X} - \mathbf{A}\mathbf{X}^*\|_2^2.$$

SNP uses a linear projection $\mathbf{M}$ to project inputs onto an orthogonal subspace. We may rewrite

$$\arg\min_{\mathbf{A}}\|\mathbf{X} - \mathbf{A}\mathbf{X}^*\|_2^2 = \arg\min_{\mathbf{A}}\|\mathbf{X} - \mathbf{A}(\mathbf{M}_*^*)^{-1}\hat{\mathbf{X}}^*\|_2^2.$$

Solving for $\mathbf{A}(\mathbf{M}_*^*)^{-1}$ by traditional LLS, we get

$$\mathbf{A}(\mathbf{M}_*^*)^{-1} = \mathbf{X}(\hat{\mathbf{X}}^*)^{\top}(\hat{\mathbf{X}}^*(\hat{\mathbf{X}}^*)^{\top})^{-1}.$$

We observe that our latent variables $\hat{\mathbf{X}}$ are orthogonal by definition of SNP. Therefore, we may rewrite the equation using $\hat{\mathbf{X}}^*(\hat{\mathbf{X}}^*)^{\top} = \mathbf{D}_*^*$. Note that pruning the row dimension of the left, and column dimension of the right matrix in a product may be expressed by pruning its product in both dimensions.

$$\mathbf{A}(\mathbf{M}_*^*)^{-1} = \mathbf{X}(\hat{\mathbf{X}}^*)^{\top}(\mathbf{D}_*^*)^{-1} = \mathbf{M}^{-1}\hat{\mathbf{X}}(\hat{\mathbf{X}}^*)^{\top}(\mathbf{D}_*^*)^{-1},$$

where we used the definition of our GS transformation to obtain the last equation. Given that $\hat{\mathbf{X}}(\hat{\mathbf{X}}^*)^{\top} = \mathbf{D}_*$,

$$\mathbf{A}(\mathbf{M}_*^*)^{-1} = \mathbf{M}^{-1}\mathbf{D}_*(\mathbf{D}_*^*)^{-1} = \mathbf{M}^{-1}\mathbf{I}_* = (\mathbf{M}^{-1})_*,$$

where $\mathbf{I}$ is the identity matrix. We find that a matrix a matrix $\mathbf{A} = (\mathbf{M}^{-1})_*\mathbf{M}_*^*$ optimally minimizes the squared approximation error of the unpruned inputs and the approximation from the pruned inputs. Notably, the recovery matrix $\mathbf{A}$ is precisely equivalent to the product of the pruned subspace transformation matrices as outlined earlier. Consequently, instead of computing the LLS approximation, the same recovery matrix can be obtained by employing the SNP method.

With this equivalence established, we demonstrate that our method optimally approximates the original inputs from their pruned counterparts in a linear manner, thereby proving the efficacy of our approach in reducing the error induced by pruning. Moreover, it underscores that the approximation process is independent of the specific ordering of units within the pruned and retained sets – a result that is not immediately apparent from our approach of pruning within a ranked subspace.

In comparison to existing literature, our method focuses on recovering the inputs, whereas approaches by Mariet & Sra (2015) and He et al. (2017) employ LLS to derive a new weight tensor. Despite this distinction, the resulting reparameterized weights are fundamentally equivalent. If we recast the LLS problem in their framework as one of approximating the layer outputs from pruned inputs, we seek to optimize a novel weight matrix $\hat{\mathbf{W}}$ through the following minimization:

$$\arg\min_{\hat{\mathbf{W}}}\|\mathbf{Y} - \hat{\mathbf{W}}\mathbf{X}^*\|_2^2.$$

By decomposing $\hat{\mathbf{W}} = \mathbf{W}\mathbf{A}$, we may equivalently optimize for $\mathbf{A}$ in:

$$\arg\min_{\mathbf{A}}\|\mathbf{W}\mathbf{X} - \mathbf{W}\mathbf{A}\mathbf{X}^*\|_2^2.$$

The remainder of the proof follows trivially from our proof above, with the only difference being the inclusion of the weight term. The resulting recovery matrix $\mathbf{A}$ is identical to that obtained before, thereby demonstrating that these methods for recovery are equivalent.

By revealing these insights, we further solidify the robustness and generality of our method.

## B  Algorithm including permutations

Below is described the same steps as outlined in the main text, but now including a permutation matrix by which the data could be re-organised prior to subspace node pruning. Note that the permutation matrix is defined based upon any additional importance scoring which is combined with our method.

We adapt Algorithm 1 to choose a particular importance score, by simply permuting the input features in $\mathbf{X}_l$, such that the features are sorted from most- to least important. In order to keep the adjacent layers unaffected, we unpermute the subspace transformation and its inverse after pruning and prior to the weight matrix multiplication. See Algorithm 2 for the algorithm description.

---

**Algorithm 2:** Layer-wise subspace node pruning with permutation

**Input:** Data $\mathbf{X}_l$, Permutation Matrix $\mathbf{P}_l$, Weights $\mathbf{W}_l$, Number of units to prune $n$

**Output:** Pruned weights $\hat{\mathbf{W}}_l$

$\mathbf{C}_l = (\mathbf{P}_l\mathbf{X}_l)(\mathbf{P}_l\mathbf{X}_l)^T$  $\qquad\qquad$ ▷ Compute dot-product between (permuted) input feature vectors

$\mathbf{M}_l^{-1}, \mathbf{D}_l = \text{LDL}(\mathbf{C}_l)$ $\qquad\qquad\qquad\qquad\qquad\qquad\qquad\qquad$ ▷ Decompose matrix $\mathbf{C}$

$\hat{\mathbf{W}}_l = \mathbf{W}_l\mathbf{P}_l^{-1}(\mathbf{M}_l^{-1})_{:,:n}(\mathbf{M}_l)_{:n,:n}(\mathbf{P}_l)_{:n,*}$ $\qquad$ ▷ Prune $\mathbf{M}$ and $\mathbf{M}^{-1}$ (leading to pruned $\mathbf{W}_l$)

$\qquad\qquad\qquad\qquad\qquad\qquad\qquad\qquad\qquad\qquad\qquad\qquad$ ▷ Note: * indicates non-zero columns only

**Return:** $\hat{\mathbf{W}}_l$

---

## C  Reconstruction-aware importance scoring

In the main text, we have discussed a reordering strategy based on the diagonal elements of the unnormalized ZCA matrix that describe the latent variances in an orthogonal subspace. The orthogonal subspace ensures that we only estimate importance from information that we cannot recover via LLS and our reconstruction method. Here, we show that instead of a ranking based on the variances, we can use a wider range of importance scoring methods while disregarding recoverable information. In particular, we demonstrate how the simple summed absolute weights (SAW) importance measure possibly benefits from this idea.

First, we need to note that we cannot reconstruct any activity from unit inputs that are already orthogonal. If all inputs were orthogonal, $\mathbf{X}_l = \hat{\mathbf{X}}_l$ and subsequently $\mathbf{M}_l = \mathbf{I}_l$. Therefore, if we assumed all inputs to be orthogonal, we could prune with minimal impact on the subsequent layer and consequently estimate importance without considering the unit activity that we may recover.

In the following, we assume that any weight tensor may be decomposed into an input orthonormalization matrix and a transformation of the latent variables within that orthonormal subspace. We have $\mathbf{W}_l = \tilde{\mathbf{W}}_l(\mathbf{X}_l\mathbf{X}_l^\top)^{-\frac{1}{2}} = \tilde{\mathbf{W}}_l\mathbf{\Sigma}_l^{-\frac{1}{2}}$, with orthonormalization matrix $\mathbf{\Sigma}_l^{-\frac{1}{2}}$ obtained by ZCA and the underlying weight transformation to the latent variables $\tilde{\mathbf{W}}_l$ (Ahmad, 2024). Therefore, given a linear layer parameterized as $\mathbf{Y}_{l+1} = \mathbf{W}_l\mathbf{X}_l$, we equivalently write as $\mathbf{Y}_{l+1} = \tilde{\mathbf{W}}_l\mathbf{\Sigma}_l^{-\frac{1}{2}}\mathbf{X}_l$. A look at the cross-correlations of the unit outputs

$$\mathbf{Y}_{l+1}\mathbf{Y}_{l+1}^\top = \tilde{\mathbf{W}}_l\mathbf{\Sigma}_l^{-\frac{1}{2}}\mathbf{X}_l\mathbf{X}_l^\top(\mathbf{\Sigma}_l^{-\frac{1}{2}})^\top\tilde{\mathbf{W}}_l^\top = \tilde{\mathbf{W}}_l\tilde{\mathbf{W}}_l^\top,$$

reveals that they are now fully defined by the transformation $\tilde{\mathbf{W}}_l$. Hence, our proposal to measure the SAW measure of $\tilde{\mathbf{W}}_l = \mathbf{W}\mathbf{\Sigma}_l^{\frac{1}{2}}$, instead of the weights.

Now, we compare the efficacy by pruning VGG networks without retraining. We refer to this novel method as SNP-SAW-tilde and compare its performance to the original SNP-SAW before retraining.

Figure 6 shows that this novel method of importance scoring cannot keep up with the good performance of SNP-SAW.

Since this measure is still a very novel measure, it requires further analysis, including retraining. Compared to SNP-SAW, our novel scoring first removes the scaling inherent to each unit, such that all latent variables have unit scaling. Therefore, the scaling measured by SAW is independent from the initial scaling of the inputs. This is different from SNP-SAW and SNP-ZCA that both perform

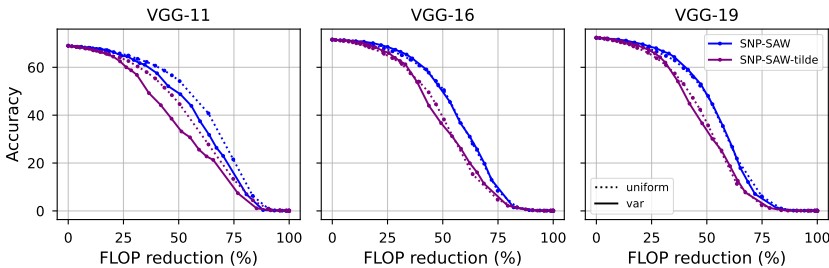

Figure 6: Similar to Figure 4, we compare SNP-SAW-tilde against SNP-SAW on VGG-16 without retraining.

superior to all our other methods after retraining. We hypothesize that a measure of this unit scaling may increase the performance. Furthermore, ZCA is a way of all-to-all orthonormalization. Therefore, this overestimates ability to reconstruct upon pruning as pruned units no longer contribute to the orthonormalization. This overestimation may be, albeit very costly, circumvented by recomputing the measure after each pruning step. Note that this equally applies to our SNP-ZCA measure.

While this measure has not yet been proven to improve upon other measures evaluated herein, it is certainly interesting for future work.

## D  HYPERPARAMETER OVERVIEW

The retraining recipe is a significant contributing factor for network performance after retraining (Wang et al., 2023a). In order to keep fair comparisons, we re-implemented a few important baselines in the literature on VGG-16 and compared their performance under the same retraining recipe. For VGG-16 we empirically found that our models performed best under the initial training recipe for the networks by (Paszke et al., 2019).

For ResNet-50, we used the recipe by (Wang et al., 2023a;b) to ensure a comparison to the novel methods under the same retraining recipe. Table 3 shows these recipes we used.

Table 3: Hyperparameter overview for retraining VGG-16 and ResNet-50 on ImageNet with PyTorch.

| Hyperparameter | VGG-16 | ResNet-50 |
|---|---|---|
| Optimizer | | SGD with momentum |
| Learning Rate | | 0.01 |
| Momentum | | 0.9 |
| Weight Decay (L2 Regularization) | | $5 \times 10^{-4}$ |
| Batch Size | | 256 |
| Number of Epochs | | 90 |
| Learning Rate Scheduler | StepLR | MultiStepLR |
| Steps at Epochs | 30 / 60 | 30 / 60 / 75 |
| Learning Rate Decay Factor | | 0.1 |
| Data Augmentation | | RandomResizedCrop, RandomHorizontalFlip |
| Normalization | | Mean: [0.485, 0.456, 0.406] Std: [0.229, 0.224, 0.225] |
| Loss Function | | CrossEntropyLoss |

## E  PRUNING RATIOS

For our analysis on VGG networks without retraining, we incremented the global pruning ratios in steps of 0.01 until 0.1, 0.025 until 0.3 and then used steps of 0.1. For ThiNet, we used a stepsize of 0.1 throughout.

For our retraining analysis, the we used the following pruning ratios for uniform pruning: [0.1, 0.2, 0.3, 0.4, 0.5].

Table 4 shows the pruning ratios for the variance and PCA heuristics. We used the ratio with the closest FLOP count compared to the uniform pruning ratios. We similarly evaluated the FLOP count of pruned ResNet-50 models. Here we incremented the pruning ratio by 0.01 until the desired FLOP count was reached.

Table 4: Pruning ratios for VGG-16 retraining experiments.

| Method | Pruning ratios |
|--------|----------------|
| SAW    | [0.04, 0.1, 0.175, 0.25, 0.3] |
| ZCA    | [0.03, 0.8, 0.125, 0.2, 0.275] |
| C      | [0.03, 0.08, 0.15, 0.225, 0.3] |
| PFA-EN | [0.01, 0.04, 0.06, 0.1, 0.15] |

## F    RESNET PRUNING

To prune networks with skip connections, we adapt the Dependency Graph (DepGraph) framework (Fang et al., 2023). DepGraph groups layers so that when a node is pruned in one layer, the corresponding nodes in related layers are pruned as well. In ResNets, information flows through two parallel branches: the residual branch and the main branch. These branches merge by summing their outputs elementwise, meaning both pathways must have the same output dimensionality to fully take advantage of pruning a node. DepGraph ensures that when a node is pruned, the corresponding input and output connections are also removed in computationally adjacent layers. Consequently, it is not clear which layer of a group to use for importance scoring. In this work, we prune nodes based upon the input activity. Naturally, this extends to scoring group-wise input activities for our unnormalized-ZCA importance and the global variance-based pruning cutoffs.

