# OpenReview forum: "Subspace Node Pruning"
_ICLR.cc/2025/Conference — ICLR 2025 Conference Withdrawn Submission_

### Official Review · Reviewer_sx8j · 2024-11-02

**Soundness:** 2
**Presentation:** 1
**Contribution:** 2
**Rating:** 3
**Confidence:** 4

**Summary:**

This paper proposes a method to prune/discard nodes (or units) from a neural network. The authors propose to project intermediate representations to a subspace such that the subspace makes the nodes or units orthogonal. Moreover, the authors provide a method to re-order nodes before orthogonalization. They also provide a global pruning saliency scheme based on the orthogonalization used. Empirical investigation is performed on the ImageNet dataset with VGGs and ResNets with and without retraining. The results indicate that the method performs reasonably well agains the selected baselines. The ZCA reordering scheme does not seem to be much useful.

**Strengths:**

S1. The problem of structured pruning, with and without pruning is hard and important.

S2. The empirical results on the selected baselines seem encouraging.

**Weaknesses:**

W1. [Writing] The paper is poorly written, and often imprecise. It is difficult to comprehend critical sections. Citing a few instances now
1.  The writing flow needs to improved in section 2.
  - What is meant by least possible impact on dynamics [line 141]?
  - Why is it important for the subspace projections to be orthogonal?
  - Line 159. "..we wish for the final dot product between each pair of vectors.." Which vectors (row vectors or column vectors of X? Isn't the orthogonalisation dependent on the samples used? How do you tackle this challenge?
  - The justification for requiring a lower-triangular matrix seems non-coherent.
2. Line 204 is supposed to be an importance score.
  - Is the matrix a diagonal matrix per definition of diag in line 161?
  - How is this different from importance in line 279? what is the relation between the two?
3. The title of the manuscript is not quite indicative of the manuscript.

W2. [Experiments] The empirical evidence is not sufficient for the following reasons.
1. More recent baselines have been left out which are quite strong; such as [a], and [b].
2. Experiments (especially those without retraining) do not compare against other weight re-compensation based methods.
3. How sensitive is the method to the number of samples (s) selected? And how sensitive is the method to the samples used.
4. It is not clear how computationally intensive is the method.
5. Inference time improvements upon structured pruning aren't measured, which is a stated motivation for this problem.
6. Experiments are not demonstrated on the CIFAR-10 and CIFAR-100 datasets which is a minimal expectation within the pruning literature. Moreover, effects of this pruning method on other architectures, such as densenets and mobilenets, have not been studied.
7. Why are SNP-SAW results not reported in Table 2?

W3. [Unaddressed items] Please refer to the questions below.


Some typos:
- Line 206. Do you mean Number of nodes to "keep" n?

[a] Structural Pruning via Latency-Saliency Knapsack. Shen et al. NeurIPS 2022

[b] DFPC: Data flow driven pruning of coupled channels without data. Narshana et al. ICLR 2023

**Questions:**

Q1. As pointed in appendix A, if fundamentally, your weight re-compensation scheme is similar to that derived from LLS, then what is the novel insights are your bringing on the table?

Q2. How do you extend the analysis of your work from feed-forward networks to CNNs? How computationally expensive is this method for CNNs?

Q3. In Appendix F, it is not clear how you compute grouped/joint importances/saliencies for ResNets. Natural extension of this method can be done in many ways. But, what do you do precisely?

---

### Official Review · Reviewer_D6TV · 2024-11-03

**Soundness:** 2
**Presentation:** 2
**Contribution:** 2
**Rating:** 3
**Confidence:** 4

**Summary:**

The paper proposes a pruning approach for reducing the effective weights in a neural network by calculating the effective subspaces of weights in a network and using an importance scoring to ensure the correct ordering of the pruning approach. They primarily show this approach on VGG and further on ResNet.

**Strengths:**

The paper presents their approach in a clear fashion and layout the theory behind their approach nicely. The motivation and why this works is also clearly shown. The experiments show an improvement, and the overall story is well tied together.

**Weaknesses:**

There's no proof of generalization to general networks or different architectures like the transformers, it is also not shown the sensitivity of this approach to language vs vision. Overall the paper seemed rushed and not well structured.

Also a side note on formatting and section distribution: This seems very different from general papers submitted, and while it doesn't take away from the content, the lack of formal formatting and some inter-spread typos highlight that this was rushed.

**Questions:**

* Needs further exploration on language and larger diverse datasets

---

### Official Review · Reviewer_Yva5 · 2024-11-04

**Soundness:** 3
**Presentation:** 3
**Contribution:** 2
**Rating:** 5
**Confidence:** 4

**Summary:**

The paper proposes an orthogonalization-based pruning strategy. Additionally, they use importance score-based pre-ordering for the orthogonalization, which provides a simple layerwise pruning ratio. The proposed importance score-based ordering performs better than existing importance score methods. The approach is tested on vgg and resnet architectures on imagenet classification.

**Strengths:**

1. The GS-based orthogonalization for node pruning is neat and the importance score-based pre-ordering makes sense.
2. Experiments show the benefit of the proposed pre-ordering strategy.
3. The results are better than the compared method albeit marginally, with and without retraining.

**Weaknesses:**

1. The improvements are marginal and not tested exhaustively on the latest architectures or other relevant pruning strategies.
2. Related to the above, low-rank pruning must be a baseline, given the similarity to the proposed method.
3. The novelty is limited unless there is a significant improvement over during SVD and low-rank-based pruning. Also, it is not clear which part of the method provides the most benefit (orthogonalization or the importance score).

**Questions:**

1. Could you clarify the benefits of the proposed method over SVD-based low-rank pruning?
2. Could it be extended to transformers?

---

### Official Review · Reviewer_Bd72 · 2024-11-12

**Soundness:** 2
**Presentation:** 2
**Contribution:** 2
**Rating:** 3
**Confidence:** 5

**Summary:**

The paper proposes a post-training pruning procedure for neural networks.  This procedure proceeds layer-wise, removing nodes whose activity is well explained by activity of other nodes in that layer.  To implement this heuristic, similarity of activation patterns across a dataset is used to compute an orthogonal basis for a subspace capturing node activity.  In finding this basis, a variant of ZCA is used to first rank nodes by importance, defining an ordering for Gram-Schmidt orthogonalization.  The resulting ordering and lower-triangular transformation matrix permit explicit pruning of the least important original nodes.  A global cutoff on variance in activity patterns calibrates pruning across all layers.

**Strengths:**

The approach appears well-motivated.

Global calibration of the pruning order of nodes across all layers is an attractive property.

Experiments demonstrate competitive performance compared to alternative pruning approaches on VGG networks (VGG-11, VGG-16, VGG-19), and ResNet.

**Weaknesses:**

The standard design of VGG networks may make them artificially good candidates for pruning.  Specifically, the first fully-connected (FC) layer of VGG learns weights that take a 7x7x512 feature tensor to a 4096-dimensional vector; this involves 7*7*512*4096 = 102.76 million parameters just for that single layer.  This is an incredibly inefficient design as an entire VGG-19 network has only 144 million parameters total.  Modern CNNs do more gradual reduction of spatial size (e.g., 8x8 to 4x4 to 2x2 to 1x1 instead of 7x7 directly to 1x1) and more gradual changes in feature dimensionality in order to avoid allocating an inordinate number of parameters to a single layer.

An consequence is that VGG is probably trivial to prune, especially for a method that globally prunes the network: simply remove parameters from the unnecessarily large 102M parameter layer.  Unfortunately, this also means results on VGG are not representative of how a pruning method might perform on other, more reasonably parameterized neural networks.  Thus, Table 1 (results on VGG-16) cannot be taken as representative of performance in general.

Results on ResNet-50 (Table 2) show both pruned accuracy and speedup similar to many competing methods; the proposed SNP-ZCA is not even the best in accuracy.

Missing is experimental validation beyond the VGG and ResNet CNNs.  Pruning is also highly relevant for networks with different components, such as transformer layers, and across a more diverse range of tasks, such as LLMs and generative models (e.g., GANs, diffusion).  I think a far more diverse experimental regime is needed to demonstrate the viability of the proposed approach.

**Questions:**

Additional experimental results across a diverse set of neural network architectures and tasks are needed in order to validate the proposed pruning method.  Please address this.

---

### Note · Authors · 2024-11-25

**Comment:**

We thank the reviewers for their input. It is clear from the reviews that a number of aspects of our work were unclear, therefore we endeavour to update our work in these respects. Furthermore, it appears that there is a desire for application of such pruning methods to other architectures and we shall investigate architectures that have been suggested. Best of wishes, (Authors).

**Withdrawal Confirmation:**

I have read and agree with the venue's withdrawal policy on behalf of myself and my co-authors.